# Integrating Proteomics and Lipidomics for Evaluating the Risk of Breast Cancer Progression: A Pilot Study

**DOI:** 10.3390/biomedicines11071786

**Published:** 2023-06-22

**Authors:** Natalia L. Starodubtseva, Alisa O. Tokareva, Valeriy V. Rodionov, Alexander G. Brzhozovskiy, Anna E. Bugrova, Vitaliy V. Chagovets, Vlada V. Kometova, Evgenii N. Kukaev, Nelson C. Soares, Grigoriy I. Kovalev, Alexey S. Kononikhin, Vladimir E. Frankevich, Evgeny N. Nikolaev, Gennady T. Sukhikh

**Affiliations:** 1V.I. Kulakov National Medical Research Center of Obstetrics, Gynecology, and Perinatology, Ministry of Health of Russia, 117997 Moscow, Russia; aurum19@mail.ru (N.L.S.); alisa.tokareva@phystech.edu (A.O.T.); v_rodionov@oparina4.ru (V.V.R.); agb.imbp@gmail.com (A.G.B.); anna.bugrova@gmail.com (A.E.B.); v_chagovets@oparina4.ru (V.V.C.); v_kometova@oparina4.ru (V.V.K.); e_kukaev@oparina4.ru (E.N.K.); vfrankevich@gmail.com (V.E.F.); g_sukhikh@oparina4.ru (G.T.S.); 2Department of Chemical Physics, Moscow Institute of Physics and Technology, 141700 Moscow, Russia; 3Laboratory of Omics Technologies and Big Data for Personalized Medicine and Health, Skolkovo Institute of Science and Technology, 121205 Moscow, Russia; g.kovalev@skoltech.ru; 4Emanuel Institute of Biochemical Physics, Russian Academy of Sciences, 119334 Moscow, Russia; 5V.L. Talrose Institute for Energy Problems of Chemical Physics, N.N. Semenov Federal Research Center for Chemical Physics, Russian Academy of Sciences, 119334 Moscow, Russia; 6Department of Medicinal Chemistry, College of Pharmacy, University of Sharjah, Sharjah 27272, United Arab Emirates; nsoares@sharjah.ac.ae; 7Laboratory of Translational Medicine, Siberian State Medical University, 634050 Tomsk, Russia; 8Center for Molecular and Cellular Biology, Skolkovo Institute of Science and Technology, 121205 Moscow, Russia

**Keywords:** breast cancer, metastasis, proteomics, lipidomics, mass spectrometry, serum, sentinel lymph nodes

## Abstract

Metastasis is a serious and often life-threatening condition, representing the leading cause of death among women with breast cancer (BC). Although the current clinical classification of BC is well-established, the addition of minimally invasive laboratory tests based on peripheral blood biomarkers that reflect pathological changes in the body is of utmost importance. In the current study, the serum proteome and lipidome profiles for 50 BC patients with (25) and without (25) metastasis were studied. Targeted proteomic analysis for concertation measurements of 125 proteins in the serum was performed via liquid chromatography–multiple reaction monitoring mass spectrometry (LC–MRM MS) using the BAK 125 kit (MRM Proteomics Inc., Victoria, BC, Canada). Untargeted label-free lipidomic analysis was performed using liquid chromatography coupled to tandem mass-spectrometry (LC–MS/MS), in both positive and negative ion modes. Finally, 87 serum proteins and 295 lipids were quantified and showed a moderate correlation with tumor grade, histological and biological subtypes, and the number of lymph node metastases. Two highly accurate classifiers that enabled distinguishing between metastatic and non-metastatic BC were developed based on proteomic (accuracy 90%) and lipidomic (accuracy 80%) features. The best classifier (91% sensitivity, 89% specificity, AUC = 0.92) for BC metastasis diagnostics was based on logistic regression and the serum levels of 11 proteins: alpha-2-macroglobulin, coagulation factor XII, adiponectin, leucine-rich alpha-2-glycoprotein, alpha-2-HS-glycoprotein, Ig mu chain C region, apolipoprotein C-IV, carbonic anhydrase 1, apolipoprotein A-II, apolipoprotein C-II and alpha-1-acid glycoprotein 1.

## 1. Introduction

Breast cancer (BC) is the most pressing issue in modern oncology, primarily due to its high incidence. According to GLOBOCAN, a joint project of the World Health Organization (WHO) and the International Agency for Cancer Research (IARC), there were 2.3 million new cases of BC registered worldwide in 2020 [1]. BC affects approximately 25% of women, making it the most common cancer among them. In the Russian Federation, breast cancer surpasses other malignant neoplasms, with 75,062 new cases detected in 2020 [2]. The exact cause of BC is unknown, but several factors can increase the risk of developing it, including age, family history of breast cancer, inherited gene mutations (such as BRCA1 and BRCA2), obesity, hormonal imbalances, and exposure to certain environmental factors [3].

While some forms of BC remain non-invasive and confined to breast tissue, others are invasive and capable of metastasizing to other parts of the body. BC metastasis occurs when cancer cells from the primary tumor break away and travel through the bloodstream or the lymphatic system to distant locations, primarily the bones, liver, lungs, or brain [4]. Metastatic BC is a serious and often life-threatening condition, representing the leading cause of death among women with breast cancer [5]. BC metastasis is a complex process involving multiple stages and factors. It begins with the invasion of cancer cells into the surrounding tissue, followed by their entry into blood vessels or lymphatic vessels (extravasation). Once in the circulation, cancer cells can reach other body parts and extravagate, exiting the vessels to invade new tissue [6]. This process can be influenced by various factors, including genetic mutations, the tumor microenvironment and the immune system [7].

Approximately 5% of women newly diagnosed with BC are already at an advanced metastatic stage [1]. Moreover, 20–30% of patients subsequently develop metastases. The diagnosis of BC metastasis typically involves a combination of imaging tests such as computed tomography (CT), magnetic resonance imaging (MRI), bone scans and positron emission tomography (PET) scans, along with biopsies or aspiration of suspected metastatic sites [8]. Blood tests, such as CA 15–3 and CEA, may be used to monitor tumor progression and treatment response [9]. Sentinel lymph node biopsy (SLNB) is a common surgical procedure used to determine if BC has spread to the lymph nodes [10]. By identifying the sentinel lymph node, which is the initial lymph node likely to be affected by cancer spread, SLNB helps avoid an unnecessary removal of healthy lymph nodes [11]. However, SLNB is not 100% accurate, and false-negative results can occur, meaning that cancer may be present in other lymph nodes even if the sentinel lymph node tests negative for cancer. The false-negative rate for SLNB ranges from 2% to 14%, depending on factors such as tumor size, location, and the surgeon’s experience [12].

Blood proteomic studies involve analyzing proteins in blood samples to identify potential biomarkers for the diagnosis of BC [13,14]. Proteins are essential molecules that perform various functions in the body, and their levels and patterns can be altered in the presence of cancer. In blood proteomic studies, researchers use techniques such as mass spectrometry and protein microarrays to analyze a large number of proteins in blood samples [15]. A review of blood-based proteomics in BC has highlighted the diagnostic potential of 51 blood proteins, with CAMK2A, APOA2 and TNFA found to be overexpressed at advanced stages of the disease [16]. It is important to note that different studies may identify different protein markers, and the clinical utility of these markers may vary depending on the population being studied and the methods used for detection and measurement [16]. Further research is needed to validate these findings and develop clinically useful diagnostic and prognostic tests based on these protein markers. The first attempt at developing a diagnostic model based on the quantitative analysis of three serum proteins (apolipoprotein C-I (APOC1), carbonic anhydrase I (CA1) and neural cell adhesion molecule L1-like protein (CHL1)) was made by the Eun-Shin Lee group in 2015, and it was validated in a prospective multicenter trial [17,18,19].

Lipids perform various functions in the organism and cells, including energy storage, signaling and building blocks for membranes [20]. Malignantly transformed cells alter lipid metabolism to promote their proliferation, migration and invasion [21]. Warburg aerobic glycolysis, abnormal amino acid and lipid metabolism are the metabolic markers of cancer cells [22]. Phosphatidylserine (PS), phosphatidylinositol (PI), phosphatidylcholine (PC) and triacylglycerol (TAG) show a positive correlation with the metastatic potential in BC cells [23,24]. Additionally, a high expression of sphingomyelin (SM) synthase and aberrant SM membrane contents have been detected [25]. Specific alterations in extracellular lipid levels (plasma, urine) have been linked to BC [16,26]. However, there are few studies on the blood lipidome in metastatic BC [26,27,28]. A high level of low-density lipoproteins (LDL > 110 mg/dL) in the blood increases the risk of lymph node metastasis [26,27,28,29]. Serum lysophosphatidylcholine (LysoPC) 22:1 and phosphatidylserine (PS) 22:0/0:0 are highly associated with the 5-year survival rate in triple-negative BC (TNBC), which is the worst biological subtype of BC [28].

Therefore, blood-based multi-omics techniques have emerged as a promising approach for the diagnosis of metastatic BC. In this study, serum proteomic and lipidomic profiles were studied to assess the risk of metastasis in BC. Targeted proteomic analysis via multiple reaction monitoring mass spectrometry (MRM MS) with internal stable isotope standards (SIS) was performed to quantify 125 proteins in 50 non-depleted serum samples. Untargeted label-free lipidomic analysis was performed for the same set of samples using liquid chromatography coupled to tandem mass spectrometry (LC-MS/MS) in both positive and negative ion modes. Finally, the highly accurate classifier that enables distinguishing between metastatic and non-metastatic BC was developed.

## 2. Materials and Methods

### 2.1. Characteristics of Study Population

This study included 50 female patients with BC from the National Medical Research Center of Obstetrics, Gynecology, and Perinatology named after V.I. Kulakov in Moscow, Russia. Out of these patients, 25 had metastases to regional lymph nodes, while the remaining 25 were free of metastasis. Prior to participating in the experiment, all subjects provided written informed consent, and the study received approval from the institutional ethics committee.

All patients underwent a standard general clinical examination, which included a detailed medical history, physical examination, blood and urine tests, assessment of histological type, tumor localization, multifocality and tumor grade. Tissue immunohistochemistry was also conducted to determine the presence of estrogen receptors (ER), progesterone receptors (PR), human epidermal growth factor receptor-2 (HER2) and the proliferation factor Ki67 (refer to Table 1 for details). Appendix A contains additional information on the patient data.

### 2.2. Sample Collection

Blood serum samples were collected in Serum Z/9 tubes (Monovette, Sarstedt, Germany) prior to surgical procedures and any therapy. The collected samples were then subjected to centrifugation at 300× *g* and 4 °C for 20 min. Subsequently, the resulting supernatant was subjected to a second round of centrifugation at 12,000× *g* for 10 min. Finally, the processed serum samples were frozen and stored at −80 °C for further analysis.

### 2.3. Targeted Proteomics via LC–MRM MS

Targeted quantitative mass spectrometry (MS) analysis of the serum proteome of BC patients was performed using a BAK 125 kit (MRM Proteomics Inc., Montreal, QC, Canada). The kit included a stable-isotope-labeled internal standard (SIS) and natural (NAT) synthetic proteotypic peptides for measuring the concentration of corresponding proteins in the blood. All MRM assays in the BAK 125 kit were characterized following the guidelines of the Clinical Proteomic Tumor Analysis Consortium (CPTAC) [29].

Sample preparation and subsequent LC–MS analysis were performed according to the manufacturer’s protocol [30]. In this study, each protein was quantified using a single tryptic peptide to maximize the number of proteins quantifiable in a single run. MS analysis was carried out on a QTRAP SCIEX6500+ mass spectrometer (SCIEX, Concord, ON, Canada). Skyline Quantitative Analysis software (version 20.2.0.343, University of Washington) was employed to visually examine the LC–MRM MS data.

Chromatographic peaks for the NAT and SIS peptides in the samples, as well as calibration curves and quality controls (QCs), were manually assessed for the shape and accurate integration. Calibration curves were generated using 1/x^2^-weighted linear regression and were used to calculate the peptide concentrations in the samples as fmol per μL of serum. Appendix A provides MRM transitions (Appendix A), the exemplary MRM data (from Skyline) and calibration curves (Appendix A).

### 2.4. Untargeted Lipidomics by LC–MS/MS

Lipid extracts were prepared following the modified Folch method [31]. In this experiment, 480 μL of a chloroform–methanol mixture (2/1, *v*/*v*) was added to 40 μL of a serum sample. The mixture was subjected to ultrasound for 10 min, after which 150 μL of water was added. The resulting mixture was then centrifuged for 5 min at 13,000× *g* and ambient temperature. The organic layer containing lipids was collected, subjected to vacuum drying, and subsequently re-dissolved in a mixture of 100 μL isopropanol and 100 μL acetonitrile for MS analysis.

The lipid extracts were randomized and analyzed in triplicate using the Dionex UltiMate 3000 liquid chromatograph (Thermo Scientific, Germering, Germany) coupled to the Maxis Impact qTOF analyzer with an electrospray ionization source (Bruker Daltonics, Bremen, Germany). A 3 μL sample was injected into a Zorbax XDB-C18 column (250 × 0.5 mm, 5 μm; Agilent, United States). Lipid separation was performed at a flow rate of 35 μL/min using a solvent system consisting of water–acetonitrile (40/60, *v*/*v*) with 0.1% formic acid and 10 mmol/L ammonium formate as solvent A, and isopropanol/acetonitrile/water (90/8/2, *v*/*v*/*v*) with 0.1% formic acid and 10 mmol/L ammonium formate as solvent B. The separation was achieved using a linear gradient from 30% to 95% (*v*/*v*) of solvent B over 25 min. The column temperature was maintained at 50 °C. Mass spectra were acquired in a positive ion mode in the *m*/*z* range of 400–1500, and in a negative ion mode in the *m*/*z* range of 100–1000 using the following settings: capillary voltage of 4.1 kV in a positive ion mode and 3.0 kV in a negative ion mode, spray gas pressure of 0.7 bar, drying gas flow rate of 6 L/min, and drying gas temperature of 200 °C. Tandem MS experiments were conducted with the three most intense peaks selected after a full scan in the full mass range, which were then fragmented via collisional dissociation with an energy of 35 eV. An ion exclusion time of 1 min was employed.

The raw LC–MS files were converted into the open MzXml format, containing information on the full MS, and into the ms2 format, containing information on tandem MS, using the msConvert program from the Proteowizard 3.0.9987 package [32]. Additionally, the MzMine 2.26 program [33] was utilized to isolate peaks, normalize them to the total ion current, and generate a table containing information on the ion mass, chromatographic peak area and retention time. Lipid identification was performed using LipidMatch 3.5 scripts [34], and the lipid nomenclature corresponds to LipidMaps [35]. Supplementary provides typical LC–MS data for lipids (Appendix A).

### 2.5. Statistical Analysis

The statistical data processing was conducted using RStudio (version 1.383 GNU). For quantitative data, median values (Me) and quartiles (Q1, Q3) were used for description. Qualitative data were presented as absolute values and as a percentage (%). Comparative analysis of qualitative data was performed using Fisher’s exact test and chi-square test. For comparative analysis of quantitative data, the Mann–Whitney test was used for pairwise comparisons between the groups, and the Kruskal–Wallis test was used for comparisons involving more than two groups. The significance threshold was set at 0.05. The Spearman test with a threshold of 0.05 was employed to estimate the correlation coefficients between two quantitative variables. To examine the impact of the markers of metastatic progression, tumor size and G on metabolic pathways, a hypergeometric test and degree centrality analysis using Metaboanalyst were employed [36].

Logistic-regression-based diagnostic models were created by generating new datasets that included both original features and features obtained by multiplying one original feature by another or itself. Using the product of features can include features with small univariate separation possibility in the model, but the product of them has higher separation possibility [37,38]. The addition product of the feature in the models can help create more quality models than using only the linear term [38,39]. Subsequently, potential variables of the logistic regression model were pre-selected using discriminant analysis orthogonal projections on latent structures and variable importance selection (with a threshold greater than 1). The logistic regression proceeded by the stepwise addition of variables, aiming to minimize the Akaike information criteria (AIC), while ensuring that the coefficients of markers had a non-zero probability below 0.05. The variables were then eliminated step by step, starting with the variable with the highest non-zero probability, until no coefficients of markers had a non-zero probability above 0.05 [40]. The quality of the models was evaluated using leave-one-out cross-validation, and the optimal threshold was determined by maximizing the sum of sensitivity and specificity.

## 3. Results

### 3.1. BC Metastasis Biomarkers in the Blood

This study focuses on the multi-omics search for specific markers of metastatic BC, and the correlation between protein/lipid markers and clinical data. Mass-spectrometry-based quantitative analysis was performed for 50 non-depleted serum samples for BC patients with and without metastasis. Targeted proteomic analysis for concertation measurements of 125 proteins in the serum was performed using LC–MRM MS with corresponding SIS peptides and resulted in 87 quantitatively measured proteins. Untargeted label-free lipidomic analysis was performed using LC–MS/MS in both positive and negative ion modes. As a result, 295 lipids were identified and quantified: 173 lipids in the positive ion mode and 159 lipids in the negative ion mode. For a complete list of the lipids and proteins in the samples, please refer to Appendix A.

Two proteins (coagulation factor XII and vasorin) and nine lipids demonstrated significant differences in concentration between cases with and without metastasis (Figure 1). Most of these lipids belong to the sphingomyelin and oxidized lipid classes. Furthermore, the levels of oxidized lipids increased in metastatic samples, while sphingomyelin levels decreased accordingly (Appendix A). We observed a weak positive correlation (*p* < 0.05, r > 0.3) between the number of metastases and the oxidized lipids (OxPC 16:0_22:5(OH), OxPC 18:0_18:2(OOH), OxTG 16:0_16:0_18:3(OOH), OxTG 16:0_18:0_18:3(OH)), triglyceride TG 14:0_16:0_18:1, apolipoprotein A-II, apolipoprotein(a) and coagulation factor XII. Additionally, sphingomyelins SM d18:2/16:0, SM d18:2/24:1, SM d22:0/20:3 and SM d22:0/20:4 exhibited a weak negative correlation (*p* < 0.05, r < −0.3) with the number of metastases.

Several proteins and lipids demonstrated significant changes based on the histological type of breast cancer (BC) (Appendix A). Particularly noteworthy are the differences observed in the proteome profile for lobular type BC. We discovered a connection between the serum levels of adiponectin, apolipoprotein D, beta-ala-his-dipeptidase, and plasma serine protease inhibitor with the biological subtype of BC (Figure 2A). Adiponectin and apolipoprotein D were significantly elevated in triple-negative breast cancer (TNBC). The lipidomic profile of serum revealed an increase in oxidized lipid levels in luminal A BC (Figure 2B). Furthermore, a group of seven proteins (alpha-2-macroglobulin, apolipoprotein L1, attractin, ceruloplasmin, hyaluronan-binding protein 2, inter-alpha-trypsin inhibitor heavy chain H2, thrombospondin-1 and vitronectin) showed correlations with BC grade and with each other. This trend was also observed for sphingomyelins, phosphatidylcholines and phosphatidylcholines with ether bonds (Appendix A).

The primary molecular processes associated with the changes in serum during sentinel lymph node metastasis include the biosynthesis of unsaturated fatty acids, linoleic acid metabolism, cholesterol metabolism, regulation of lipolysis in adipocytes, fatty acid biosynthesis and arachidonic acid metabolism (Appendix A). However, after applying the Benjamini–Hochberg correction, only the first three pathways remained statistically significant (FDR < 0.05).

### 3.2. Building of a Binary Classifiers for BC Metastasis Diagnosis

The prediction models for BC metastatic progression were based on logistic regressions, which use as independent variable feature values of first and second order. Variables were selected by combining variable importance projection filter (with VIP > 1 threshold) and an Akaike Information Criteria (AIC) algorithm for more stability for feature selection (see more details in Methods—Section 2.5).

The proteomic-based model for sentinel node metastasis included the following proteins: alpha-2-macroglobulin, coagulation factor XII, adiponectin, leucine-rich alpha-2-glycoprotein, alpha-2-HS-glycoprotein, Ig mu chain C region, apolipoprotein C-IV, carbonic anhydrase 1, apolipoprotein A-II, apolipoprotein C-II and alpha-1-acid glycoprotein 1 (Table 2). Coagulation factor XII and apolipoprotein A-II were found to be negatively correlated with the number of metastases in previous findings. The model demonstrated a sensitivity of 91% and specificity of 89%, with a threshold of 0.31 (Figure 3A).

The lipidomic-based diagnostic model consisted of specific lipid molecules, including triacylglycerols (OxTG) such as OxTG 16:0_18:0_18:3(OH), OxTG 18:1_18:1_18:1(Ke,OH), TG 16:0_16:1_18:1 and TG 18:1_18:1_18:2. Additionally, phosphatidylcholines (PC) PC 18:0_22:6 and PC 16:1_20:4, as well as sphingomyelins (SM) SM d18:2/24:1, were included in the model (Table 3). OxTG 16:0_18:0_18:3(OH) and SM d18:2/24:1 exhibited correlations with the number of metastases and showed significant changes in the metastatic lesions of regional lymph nodes. The sensitivity and specificity of this model were determined to be 96% and 64%, respectively, with a threshold of 0.31 (Figure 3B).

## 4. Discussion

The 5-year survival rate for breast cancer patients with localized tumors is as high as 99%. However, when the cancer spreads to nearby lymph nodes, the survival rate decreases to 86%, and further metastasis to distant parts of the body results in a survival rate of only 28% [5]. Lymph nodes serve as the primary sources of metastatic spread, and determining their metastatic status is crucial for appropriate treatment selection. However, the use of sentinel lymph node biopsy (SLNB) [41] can lead to the development of complications such as lymphatic stagnation, nerve damage, lymphedema, pain, and limited arm mobility [42].

Non-invasive methods such as ultrasound, mammography, magnetic resonance imaging (MRI), proton emission tomography (PET), and computed tomography (CT) are available, but they generally exhibit lower sensitivity compared to SLNB, and heavily rely on the expertise of the examining physician [43,44]. Although the current clinical classification of breast cancer is well-established, the addition of minimally invasive laboratory tests based on peripheral blood biomarkers that reflect pathological changes in the body is of utmost importance. These tests can provide valuable information to complement existing diagnostic approaches.

Quantitative analysis of plasma proteins is not an easy task, and an in-depth analysis of a wide panel of proteins using mass spectrometry methods requires additional efforts and should rely on well-validated reproducible protocols [29]. There are several strategies that are recommended for blood proteomics and clinical studies. One of these approaches for absolute concentration measurements of 125 proteins in non-depleted blood plasma/serum was proposed by Prof. Borcher’s team that has developed MRM proteomics [30]. In our work, we used the protocol and the commercial kit BAK 125 (MRM proteomics, Canada) which includes 125 medium-to-low abundance plasma proteins associated with cancer, CVD and other diseases. In this study, the concentration of 87 major blood serum proteins was measured and associations with breast cancer progression and metastasis, including tumor grade, biological and histological subtype, were found.

A significant increase in the level of coagulation factor XII was observed in regional metastasis. Furthermore, the concentration of this protein rises in the blood serum with an increase in the number of metastases. Coagulation factor XII actively participates in the immune response by enhancing the phagocytic activity of monocytes/macrophages [45,46]. In Todd et al.’s study, an elevated F12 level in the intercellular space was associated with an increased risk of death in breast cancer [47].

Apolipoproteins A-2 and A-1 also exhibited positive correlations with the number of metastases. Apolipoproteins play an active role in lipid metabolism and have anti-inflammatory effects, including the inhibition of neutrophils [48,49]. On the other hand, apolipoproteins C-2 and C-4 contributed to the diagnostic model with negative coefficients. Both proteins are involved in the metabolism and transportation of triglycerides as part of very-low-density lipoproteins [50,51]. A positive relationship between the level of apolipoprotein C-2 in the plasma and the remission period was found in patients with cervical cancer [52]. However, the situation is reversed in pancreatic cancer [53].

Adiponectin appeared in the diagnostic model of regional metastasis as part of two variables with opposite signs. Adiponectin, produced by adipocytes, plays a role in the regulation of energy storage. The level of adiponectin in blood serum decreases with the progression of breast cancer [54]. An addition of adiponectin to breast cancer cells leads to a reduction in the number of lipid vesicles [55]. Plasma adiponectin, along with eicosapentaenoic acid, docosahexaenoic acid, docosapentaenoic acid and arachidonic acid, demonstrates anti-inflammatory activity [56]. However, a meta-analysis by Zeping Yu et al. did not establish a statistically significant association between the blood levels of adiponectin and metastatic disease [57].

Leucine-rich alpha-2-glycoprotein (LRG), alpha-2-HS-glycoprotein and alpha-2-macroglobulin were included in the diagnostic model. LRG has been found to inhibit apoptosis [58] and, similar to alpha-acid glycoprotein, increases the risk of death [59,60]. LRG is associated with the onset of metastasis and cancer aggressiveness [61,62,63]. The group led by Chris Verathamjamras included LRG and alpha-acid glycoprotein in a model for detecting metastatic colorectal cancer [64]. Alpha-2-HS-glycoprotein promotes the growth of proliferative activity, adhesion and chemotaxis of cancer cells, thereby accelerating the development of metastatic lesions [65,66]. Alpha-2-macroglobulin activates lipogenesis processes that are necessary for the active proliferation and survival of tumor cells [67]. However, a decrease in the level of serum alpha-2-macroglobulin was observed in metastatic cancer of the rectum and colon [68,69].

In this study, untargeted label-free lipidomic analysis was performed using LC–MS/MS in both positive and negative ion modes. As a result, 295 lipids were identified and quantified: 173 lipids in the positive ion mode and 159 lipids in the negative ion mode. These lipids mainly consist of triglycerides (TG), phosphatidylcholines (PC), lysophosphatidylcholines (LPC), lysoephosphatidylethanolamines (LPE), phosphatidylinositols (PI), cholesterol esters (CE), oxylipids (OxL), plasmanyl- and plasmenyl-lipids (plasmanyl-, plasmenyl-), sphingomyelins (SM) and ceramides (Cer). Several proteins and lipids were found to be associated with breast cancer progression and metastasis, including tumor grade, biological and histological subtype.

In the cases of metastasis to the sentinel lymph nodes, a significant increase in the level of oxidized triglycerides and phosphatidylcholines was demonstrated, and this increase positively correlates with the number of metastases. This finding aligns with in vitro data, which suggest that the content of oxidized lipids in the tumor cell membrane is associated with their metastatic potential [70]. These lipids play a crucial role in the processes that support the survival of tumor cells [71,72]. Conversely, sphingomyelins exhibit the opposite trend. Acid sphingomyelase in tumor cell tissues activates cell adhesion processes and initiates metastasis [25,73]. Hui-Ming Lin et al. reported that higher plasma levels of sphingomyelins significantly worsened the prognosis in prostate cancer [74].

Dysregulated metabolic pathways in regional metastasis primarily involve the biosynthesis of unsaturated fatty acids, linoleic acid metabolism, and cholesterol metabolism (FDR < 0.05). Previous studies have demonstrated a positive correlation between LDL levels and the risk of lymph node metastasis [26,27]. LDL is responsible for transporting the majority of cholesterol in the bloodstream. Furthermore, a connection has been established between LDL and hypertriglyceridemia, obesity, metabolic syndrome and inflammation [75]. Increasing the cholesterol content in the cell membrane has been linked to cancer progression and aggressiveness [76].

Fatty acids (FAs) play a crucial role in membrane and signaling lipid synthesis, and serve as the primary energy source for cells. To meet the demands of rapid growth and metastasis, cancer cells balance anabolic and catabolic lipid metabolism. The combination of increased lipolysis and lipogenesis ensures a constant supply of fatty acids and lipids necessary for the growth and dissemination of malignant tumors [77,78]. Therefore, cancer cells’ adaptation of lipid metabolism accelerates the tumor metastatic spread [21].

Most lipids associated with BC grade and metastasis correlate with each other (Appendix A). The same trend is found for marker proteins. It confirms that proteins and lipids, proposed as markers, are indeed involved in the general pathological mechanisms of the development and metastasis of breast cancer. Nevertheless, the correlation between these proteins and lipids is non-significant in most cases, excluding vitronectin and ceruloplasmin. These proteins negatively correlate with BC grade and the plasma level of several lipid markers (for instance, PC 16:0_16:0, PC O-16:0/16:0, PC O-16:1/18:0, PC O-18:1/16:0, LPE 22:5, OxPC 18:0_20:4(OOO), SM d18:1/18:2 and SM d18:2/24:1) (Appendix A). Moreover, the protein markers of metastatic BC such as coagulation factor XII and apolipoprotein A-II, exhibit a positive link with OxTG 16:0_18:0_18:3(OH) and TG 14:0_16:0_18:1 (Appendix A). The plasma level of triglyceride TG 14:0_16:0_18:1 correlates with almost all lipids specific for metastatic processes. Apolipoprotein A-II is the second most abundant protein of the high-density lipoproteins (HDL), transporting triglycerides and phospholipids (the main sources of fatty acids for building membranes and energy metabolism of cells). HDLs have anti-atherogenic properties, modulate inflammation and blood coagulation. Cancer cells trigger thrombin generation resulting from a combination of the inherent procoagulant properties of cancer-cell-associated tissue factor, as well as of procoagulant phospholipids in the plasma microenvironment [79]. The FXII activation pathway of thrombin generation is the key to cancer-induced hypercoagulability [80]. Ceruloplasmin (CP), also known as copper oxidase, plays an important role in free-radical scavenging, exhibiting antioxidant activity, and immunity processes (a protein of acute phase). CP is related to dyslipidemia, as an effective catalyst for the oxidation of low-density lipoproteins [81]. Thus, proteins and lipids, proposed as the markers of BC progression and metastasis, participate in the cancer-related processes, in particular, chronic inflammation and hypercoagulation.

The resulting proteomic- and lipidomic-based models demonstrate the high potential for diagnosing regional metastasis with 91% sensitivity and 89% specificity (AUC = 0.92) for the protein model. The performance of the proteomic-based model was significantly better compared to machine learning models using ultrasound and MRI data [82,83], dynamic-contrast-enhanced MRI [84], MRI with genomic data [85] and ultrasound [86]. A radiomics machine learning model based on computed tomography exhibited comparable sensitivity and specificity [87].

The current study has a number of limitations. First, this is a single-center pilot study with a small cohort of 50 women. Further research is necessary with an independent patient cohort to validate the proposed serum markers of BC sentinel lymph node metastases. Secondly, the proteomic part was limited to the measurements of 125 blood serum proteins. Expanding the panel with other proteins, including BC-specific potential markers, is needed in the future to improve the quality of the developed diagnostic models. Regarding the lipidomic part, further targeted validation via LC–MRM MS with labeled standards is desirable, including an independent patient cohort. Third, this study does not include BC patients with stage IV breast cancer, because they rarely need surgery.

## 5. Conclusions

Multi-omics LC–MS-based serum analysis was performed for patients with metastatic and non-metastatic BC. As a result, 87 proteins and 295 lipids were quantitatively evaluated in the serum, and showed moderate correlations with tumor grade, the number of lymph node metastases, and the biological and histological subtypes of BC. Further pathway analysis revealed that the main molecular changes related to sentinel lymph node metastasis in BC were associated with the biosynthesis of unsaturated fatty acids, linoleic acid metabolism and cholesterol metabolism (FDR < 0.05). Finally, two highly accurate classifiers that enabled distinguishing between metastatic and non-metastatic BC were developed based on proteomic (accuracy 90%) and lipidomic (accuracy 80%) features. The best classifier (91% sensitivity, 89% specificity, AUC = 0.92) for BC metastasis diagnostics was based on a logistic regression and the serum levels of 11 proteins: alpha-2-macroglobulin, coagulation factor XII, adiponectin, leucine-rich alpha-2-glycoprotein, alpha-2-HS-glycoprotein, Ig mu chain C region, apolipoprotein C-IV, carbonic anhydrase 1, apolipoprotein A-II, apolipoprotein C-II and alpha-1-acid glycoprotein 1. The developed diagnostic models may identify patients with a risk of metastasis to regional lymph nodes, and thus help assess the volume of treatment for BC patients.

## Figures and Tables

**Figure 1 biomedicines-11-01786-f001:**
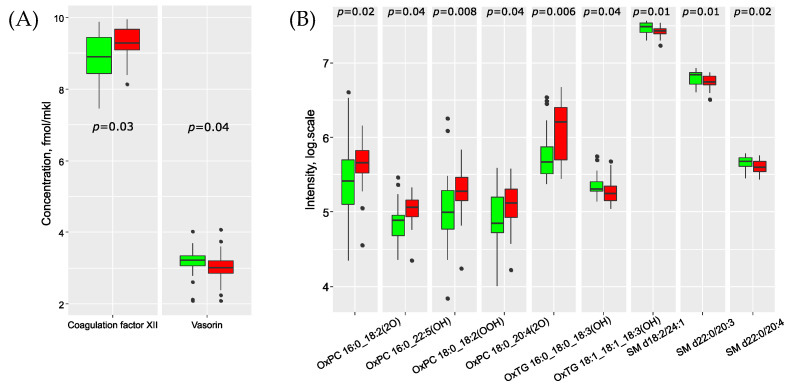
Boxplot of serum proteins (**A**) and lipids (**B**) with significantly different protein concentrations in BC patients with (red) and without (green) metastases in the sentinel lymph nodes.

**Figure 2 biomedicines-11-01786-f002:**
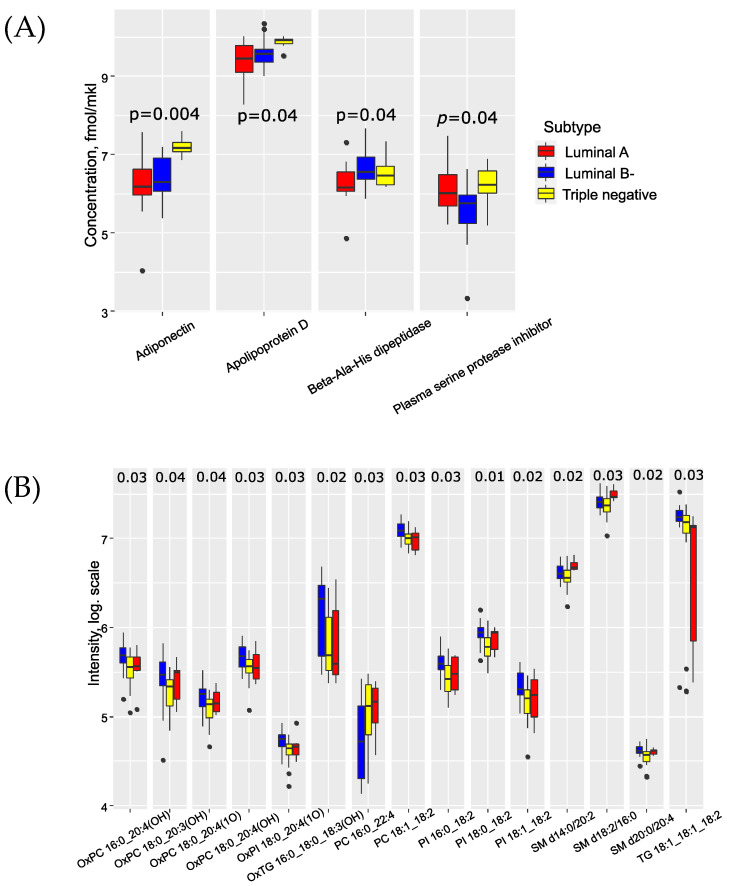
Proteins (**A**) and lipids (**B**), potential markers of BC biological subtype: blue—luminal A, yellow—luminal B−, red—triple negative).

**Figure 3 biomedicines-11-01786-f003:**
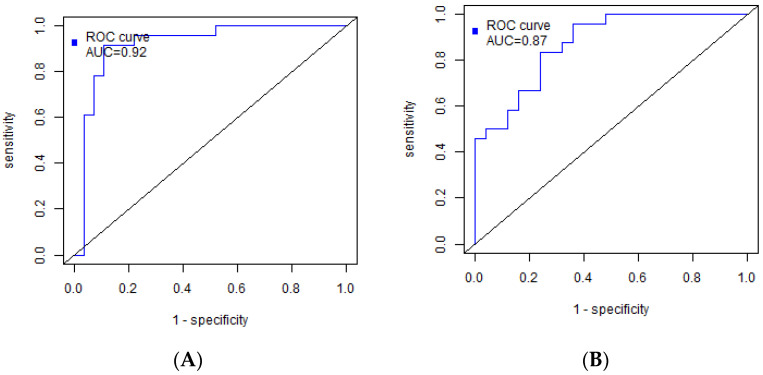
ROC curves of the models for BC sentinel node metastasis based on serum: (**A**) proteins; (**B**) lipids.

**Table 1 biomedicines-11-01786-t001:** Demographic and clinical data of BC patients.

Parameter	Metastases-Free Group (*n* = 25)	Group with Metastases (*n* = 25)	*p*-Value
Age (years)	60 (52; 63)	56 (44; 60)	0.11
Length of tumor (cm)	2.1 (1.6; 2.4)	2.4 (1.9; 3.0)	0.13
Biological subtype:			0.47
Luminal A	8 (32.0%)	10 (40.0%)	
Luminal B−	11 (44.0%)	12 (48.0%)	
Luminal B+	1 (4.0%)	0 (0.0%)	
Her2+	0 (0.0%)	1 (4.0%)	
TNBC	5 (20.0%)	2 (8.0%)	
Histological type:			0.70
Invasive ductal breast carcinoma NST	6 (24.0%)	6 (24.0%)	
Invasive lobular carcinoma NOS	5 (20.0%)	4 (16.0%)	
Invasive mixed breast cancer NST	11 (44.0%)	14 (56.0%)	
Special BCs	3 (12.0%)	1 (4.0%)	
Grade, G:			0.20
I	3 (12.0%)	1 (4.0%)	
II	13 (52.0%)	19 (76.0%)	
III	9 (36.0%)	5 (20.0%)	
Multifocality (>1 tumor):			0.66
yes	4 (16.0%)	2 (8.0%)	
no	21 (84.0%)	23 (92.0%)	
Stage:			<0.001
Ia:	13 (52.0%)	0	
Ib:	0	1 (4.0%)	
IIa	11 (44.0%)	3 (12.0%)	
IIb	1 (4.0%)	13 (52.0%)	
IIIa	0	4 (16.0%)	
IIIb	0	4 (16.0%)	
Total malignancy score (TMS)	15 (13; 16)	15 (14; 16)	0.15
Nottingham predictive index (NPI)	3.4 (3.3; 4.4)	4.7 (4.5; 5.4)	<0.001
Number of metastases to regional lymph nodes	0	2 (1; 5)	<0.001
Estrogen receptor (ER) expression:	8 (7; 8)	8 (7; 8)	0.60
Progesterone receptor (PR) expression:	7 (0; 7)	4 (2; 8)	1.00
HER2 expression:			1.00
positive	2 (8.0%)	1 (4.0%)	
negative	23 (92.0%)	24 (96.0%)	
Level of Ki67, %	28.0 (14.0; 45.0)	22.0 (15.0; 38.0)	0.96

**Table 2 biomedicines-11-01786-t002:** Variables included in the proteomic-based model for BC sentinel node metastasis. Coefficient β, confidence interval CI β, Wald criteria Z and the coefficient zero-probability *p* are provided.

Variable	β	CI β	Z	*p*
Intercept	−32.32	−75.78–−8.25	−2.00	0.04
Alpha-2-macroglobulin × Coagulation factor XII	0.69	0.33–1.33	2.87	0.004
Adiponectin × Leucine-rich alpha-2-glycoprotein	−1.71	−3.81–−0.75	−2.45	0.01
Alpha-2-HS-glycoprotein × Ig mu chain C region	0.34	0.14–0.81	2.16	0.03
Apolipoprotein C-IV × Carbonic anhydrase 1	−0.39	−0.86–−0.15	−2.33	0.02
Apolipoprotein A-II × Apolipoprotein C-II	−0.09	−0.20–−0.03	−2.41	0.02
Adiponectin × Alpha-1-acid glycoprotein 1	0.79	0.24–1.86	2.09	0.04

**Table 3 biomedicines-11-01786-t003:** Variables included in the lipid-based model for BC sentinel node metastasis. Coefficient β, confidence interval CI β, Wald criteria Z and the coefficient zero-probability *p* are provided.

Variable	β	CI β	Z	*p*
Intercept	9.52	3.47–19.42	2.43	0.02
OxTG 16:0_18:0_18:3(OH) × OxTG 18:1_18:1_18:1(Ke,OH)	1.38 × 10^−12^	4.65 × 10^−13^–3.09 × 10^−12^	2.05	0.04
SM d18:2/24:1 × TG 16:0_16:1_18:1	−2.06 × 10^−14^	−4.17 × 10^−14^–−7.73 × 10^−15^	−2.46	0.01
PC 18:0_22:6 × TG 18:1_18:1_18:2	−1.12 × 10^−14^	−2.26 × 10^−14^–−3.34 × 10^−15^	−2.41	0.02
OxTG 18:1_18:1_18:2(OOH) × PC 16:1_20:4	3.41 × 10^−14^	1.03 × 10^−14^–6.68 × 10^−14^	2.45	0.01

## Data Availability

Data are contained within the Appendix A.

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
