# Peer review of "Integrating Proteomics and Lipidomics for Evaluating the Risk of Breast Cancer Progression: A Pilot Study"

_biomedicines, 2023, doi:10.3390/biomedicines11071786_

Round 1

Reviewer 1 Report

The authors in this article study and evaluate serum proteome and lipidoma profiles in patients with breast cancer, in the presence and absence of metastases with the aim of identifying new biomarkers. For this purpose, 87 serum proteins and 295 lipids were quantified, and the authors performed correlation analyzes of the serum profile of these patients with tumor grade, histological and biological subtypes, and the number of lymph node metastases.

In general, the text presents a clear workflow logic and does not present particular grammatical errors. However, there are some errors and omissions that need to be addressed. Furthermore, to extend the research to other proteins in addition to the 87 already considered.

Author Response

The authors in this article study and evaluate serum proteome and lipidoma profiles in patients with breast cancer, in the presence and absence of metastases with the aim of identifying new biomarkers. For this purpose, 87 serum proteins and 295 lipids were quantified, and the authors performed correlation analyzes of the serum profile of these patients with tumor grade, histological and biological subtypes, and the number of lymph node metastases.

In general, the text presents a clear workflow logic and does not present particular grammatical errors. However, there are some errors and omissions that need to be addressed. Furthermore, to extend the research to other proteins in addition to the 87 already considered.

Answer. Thank you for reviewing our work and important suggestions! We agree that there are the limitations of our study including restricted protein panel.  Quantitative analysis of plasma proteins is not an easy task, and in-depth analysis of a wide panel of proteins using mass spectrometry methods requires additional efforts and should rely on well-validated reproducible protocols. There are several strategies that are recommended for blood proteomics and clinical studies. One of these approaches for absolute concentration measurements of 125 proteins in non-depleted blood plasma/serum was proposed by Prof. Borchers team developed by MRM proteomics. In our work, we used the protocol and the commercial kit BAK 125 (MRM proteomics, Canada) which includes 125 medium to low abundance plasma proteins associated with cancer, CVD, and other diseases. In this study, the concentration of 87 major blood serum proteins was measured and associations with breast cancer progression and metastasis, including tumor grade, biological and histological subtype were found.

We agree that expanding the panel with other proteins including BC specific potential markers is needed in future to improve the accuracy of current model. We added the corresponding text to Discussion section including remarks regarding limitations of current study:

 “The current study has a number of limitations. First, this is a single center pilot study with a small cohort of 50 women. Further research is necessary to validate the proposed serum markers of BC sentinel lymph nodes metastasis. Secondly, the proteomic part was limited to measurements of 125 blood proteins. Expanding the panel with other proteins including BC specific potential markers is needed in future to improve the accuracy of current model….”

Reviewer 2 Report

This study focuses on the multi-omics search for specific markers of metastatic breast cancer biomarkers. The cohort includes a metastases-free group (n=25) and a group with metastases (n=25). Overall, the manuscript is well written and the data are well presented. Results support the conclusions derived from the data. I have only a minor comment. In my opinion, validation in an independent cohort is necessary before publication.

Author Response

This study focuses on the multi-omics search for specific markers of metastatic breast cancer biomarkers. The cohort includes a metastases-free group (n=25) and a group with metastases (n=25). Overall, the manuscript is well written and the data are well presented. Results support the conclusions derived from the data. I have only a minor comment. In my opinion, validation in an independent cohort is necessary before publication.

Answer. Thank you for reviewing our work and important suggestions! We do our best to improve the manuscript. We agree that there are the limitations of our study and added the corresponding section to Discussion. It is a pilot study and the models performance was analyzed by «leave-one-out cross-validation». Indeed, further research is necessary to validate the results in an independent cohort.

This corresponding text was added in Discussion section: “The current study has a number of limitations. First, this is a single center pilot study with a small cohort of 50 women. Further research is necessary with an independent patient cohort to validate the proposed serum markers of BC sentinel lymph nodes metastasis. Secondly, the proteomic part was limited to measurements of 125 blood serum proteins. Expanding the panel with other proteins including BC specific potential markers is needed in future to improve the quality of the developed diagnostic models. Regarding the lipidomic part, further targeted validation by LC-MRM MS with labeled standards is desirable including independent patient cohort. Third, this study does not include BC patients with the IV stage of breast cancer, because they rarely need surgery.”   

Reviewer 3 Report

I have reviewed the paper "Integrating Proteomics and Lipidomics for Evaluating the Risk of Breast Cancer Progression: a pilot study" by the authors. The paper presents a multi-omics approach to identify serum biomarkers for the diagnosis of metastatic breast cancer (BC) and to explore the correlation between proteome and lipidome profiles and clinical data. The paper is well-written, clear, and informative. The methods are sound and appropriate, and the results are interesting and relevant. The paper makes a valuable contribution to the field of BC biomarker discovery and multi-omics integration.

However, I have some questions and comments that I would like the authors to address. These are:

- What are the reasons for not including stage 4 BC patients in the study? How would this affect the generalizability and applicability of the findings?

- What is the relationship between lipids and proteins in BC progression and metastasis? How do they interact or influence each other at the molecular level?

- How did the authors make the prediction models and roc analysis? What are the advantages and limitations of using logistic regression and AIC for model selection?

- How do the authors validate their findings in an independent cohort? What are the criteria for selecting the validation samples and how do they ensure their comparability with the discovery samples?

Author Response

I have reviewed the paper "Integrating Proteomics and Lipidomics for Evaluating the Risk of Breast Cancer Progression: a pilot study" by the authors. The paper presents a multi-omics approach to identify serum biomarkers for the diagnosis of metastatic breast cancer (BC) and to explore the correlation between proteome and lipidome profiles and clinical data. The paper is well-written, clear, and informative. The methods are sound and appropriate, and the results are interesting and relevant. The paper makes a valuable contribution to the field of BC biomarker discovery and multi-omics integration.

However, I have some questions and comments that I would like the authors to address. These are:

Answer: Thank you for reviewing our work and important suggestions! We did our best to improve the text. Please see below our answers step by step.

- What are the reasons for not including stage 4 BC patients in the study? How would this affect the generalizability and applicability of the findings?

Answer: Indeed, our study does not include patients with the IV stage of breast cancer, because they rarely need surgery and it’s always metastatic BC. Our aim was to developed a diagnostic models which may identify BC patients with a risk of metastasis to regional lymph nodes and thus help to assess the volume of treatment of such a patients. The following text was added to the Discussion and Conclusion sections.

- What is the relationship between lipids and proteins in BC progression and metastasis? How do they interact or influence each other at the molecular level?

Answer: Thank you for this valuable question. The following text was added to Discussion: “Most of lipids, associated with BC grade and metastasis, correlate with each other (Figure 3S and Figure 5S). The same trend is found for marker proteins. It confirms that proteins and lipids, proposed as markers, are indeed involved in the general pathological mechanisms of development and metastasis of breast cancer. Nevertheless, correlation between these proteins and lipids is non-significant in most cases, excluding vitronectin and ceruloplasmin. These proteins negatively correlate with BC grade and the plasma level of several lipid markers (for instance, PC 16:0_16:0, PC O-16:0/16:0, PC O-16:1/18:0, PC O-18:1/16:0,  LPE 22:5, OxPC 18:0_20:4(OOO), SM d18:1/18:2 and SM d18:2/24:1) (Figure 5S). Moreover, protein markers of metastatic BC, coagulation factor XII and apolipoprotein A-II, exhibit positive link with OxTG 16:0_18:0_18:3(OH) and TG 14:0_16:0_18:1 (Figure 3S). Plasma level of triglyceride TG 14:0_16:0_18:1 correlate with almost all lipids, specific for metastatic process. Apolipoprotein A-II is the second most abundant protein of the high-density lipoproteins (HDL), transporting triglycerides and phospholipids (the main sources of fatty acids for building membranes and energy metabolism of cells). HDL have anti-atherogenic properties, modulate inflammation and blood coagulation. Cancer cells trigger thrombin generation resulting from a combination of the inherent procoagulant properties of cancer cell-associated tissue factor as well as of procoagulant phospholipids in the plasma microenvironment [79]. FXII activation pathway of thrombin generation is key to cancer-induced hypercoagulability [80]. Ceruloplasmin (CP), also known as copper oxidase, plays an important role in free radical scavenging, exhibiting antioxidant activity, and in immunity processes (a protein of acute phase). CP is related to dyslipidemia, as an effective catalyst for the oxidation of low-density lipoproteins[81]. Thus, proteins and lipids, proposed as markers of BC progression and metastasis, participate in the cancer-related processes, in particular, chronic inflammation and hypercoagulation.”

- How did the authors make the prediction models and roc analysis? What are the advantages and limitations of using logistic regression and AIC for model selection?

Answer. We tried to elucidate the question concerning the prediction models and roc analysis and the advantages and limitations of using logistic regression and AIC for model selection and added corresponding text in Results - section 3.4: “Prediction models for BC metastatic progression were based on logistic regres-sions, which use as independent variable feature values of first and second order. Var-iables were selected by combining variable importance projection filter (with VIP>1 threshold) and Akaike Information Criteria (AIC) algorithm for more stability for fea-ture selection. (see more details in Methods – section 2.5).”

- How do the authors validate their findings in an independent cohort? What are the criteria for selecting the validation samples and how do they ensure their comparability with the discovery samples?

Answer. Thank you for this important question! The models performance was analyzed by «leave-one-out cross-validation». Further research is necessary to validate the results in an independent cohort. This limitation of the study was added in Discussion.

Reviewer 4 Report

Starodubtseva NL et al in this manuscript investigated proteomics and lipidomics from breast cancer patients with or without metastasis for assessing the risk of breast cancer progression. Overall, the majority of the data is convincing, and the experiments were well-designed to support the major conclusion, there are a few minor concerns that should be addressed before consideration for publication.

Comments:

1, In Results 3.1, the authors didn’t mention Fig 1B in manuscript.

2, In Line 254, “seven proteins” should be “eight proteins”.

3, The Method of building of a binary classifier is not clear.

Author Response

Starodubtseva NL et al in this manuscript investigated proteomics and lipidomics from breast cancer patients with or without metastasis for assessing the risk of breast cancer progression. Overall, the majority of the data is convincing, and the experiments were well-designed to support the major conclusion, there are a few minor concerns that should be addressed before consideration for publication.

 Answer: Thank you for reviewing our work and important suggestions! We did our best to improve the text. Please see below our answers step by step.

Comments:

  1. In Results 3.1, the authors didn’t mention Fig 1B in manuscript.

Answer. The text was corrected: “Most of these lipids belong to the sphingomyelin and oxidized lipid classes (Figure 1B).”

  1. In Line 254, “seven proteins” should be “eight proteins”.

Answer. The mistake was corrected: “Furthermore, a group of eight proteins (alpha-2-macroglobulin, apolipoprotein L1, attractin, ceruloplasmin, hyaluronan-binding protein 2, inter-alpha-trypsin inhibitor heavy chain H2, thrombospondin-1, and vitronectin) showed correlations with BC grade and with each other.”

  1. The Method of building of a binary classifier is not clear.

Answer. We tried to elucidate the question concerning the building of a binary classifier and added the corresponding text it Result - section 3.2: Prediction models for BC metastatic progression were based on logistic regressions, which use as independent variable feature values of first and second order. Variables were selected by combining variable importance projection filter (with VIP>1 threshold) and Akaike Information Criteria (AIC) algorithm for more stability for feature selection. (see more detail in Methods – section 2.5).